# Lipopolysaccharide-affinity copolymer senses the rapid motility of swarmer bacteria to trigger antimicrobial drug release

Shengtao Lu[1], Wuguo Bi[2], Quanchao Du[1], Sheetal Sinha[3,4], Xiangyang Wu[1], Arnold Subrata[1], Surajit Bhattacharjya [3], Bengang Xing[1] & Edwin K. L. Yeow[1]

An intelligent drug release system that is triggered into action upon sensing the motion of swarmer *P. mirabilis* is introduced. The rational design of the drug release system focuses on a pNIPAAm-*co*-pAEMA copolymer that prevents drug leakage in a tobramycin-loaded meso-porous silica particle by covering its surface via electrostatic attraction. The copolymer chains are also conjugated to peptide ligands YVLWKRKRKFCFI-NH$_2$ that display affinity to Gram-negative bacteria. When swarmer *P. mirabilis* cells approach and come in contact with the particle, the copolymer-YVLWKRKRKFCFI-NH$_2$ binds to the lipopolysaccharides on the outer membrane of motile *P. mirabilis* and are stripped off the particle surface when the cells move away; hence releasing tobramycin into the swarmer colony and inhibiting its expansion. The release mechanism is termed Motion-Induced Mechanical Stripping (MIMS). For swarmer *B. subtilis*, the removal of copolymers from particle surfaces via MIMS is not apparent due to poor adherence between bacteria and copolymer-YVLWKRKRKFCFI-NH$_2$ system.

[1] Division of Chemistry and Biological Chemistry, School of Physical and Mathematical Sciences, Nanyang Technological University, 21 Nanyang Link, 637371 Singapore, Singapore. [2] College of Science, Harbin Engineering University, Harbin 150080, China. [3] School of Biological Sciences, Nanyang Technological University, 60 Nanyang Drive, 637551 Singapore, Singapore. [4] Advanced Environmental Biotechnology Centre, Nanyang Environment and Water Research Institute, Nanyang Technological University, 1 Cleantech Loop, 637141 Singapore, Singapore. Correspondence and requests for materials should be addressed to E.K.L.Y. (email: edwinyeow@ntu.edu.sg)

**P**roteus mirabilis is a Gram-negative bacteria associated with urinary tract infections (UTIs) that often affect patients inserted with indwelling urethral catheters[1–4]. A unique characteristics of P. mirabilis is its ability to undergo cell differentiation into elongated and hyperflagellated swarmer cells that move rapidly across surfaces of solid media, including urethral catheter surfaces, in multicellular rafts[1–5]. The collective motion is believed to play a role in the initiation of catheter-associated UTIs (CAUTIs) by facilitating the dissemination of bacteria from infection sites on the catheter to the bladder[3,4,6,7]. The most common treatment of CAUTIs involves either regular replacements of infected catheters with new ones, administration of strong antibiotics or prevention of biofilm formation/encrustation, often with conflicting results[3,4,8]. Unfortunately, existing methods to combat CAUTIs are unable to control the release of antibiotics nor do they specifically target the initial migration of swarmer P. mirabilis from catheter surface to urinary tract.

The controlled release of drug is an important process as it allows the therapeutic agent to display its active form only at the infection site[9–11]. Previous technologies on 'on-demand' release of antimicrobial drugs encapsulated within carriers (e.g., particles, polymers, etc.) have utilized toxin/lipase secreted by bacteria to react with toxin/lipase-responsive materials on carriers to trigger drug release[11–15]. Another common stimuli-responsive method utilizes the reduced pH environment in cariogenic dental biofilms (pH ≤ 4.5) to activate drug release from pH-sensitive carriers[16,17]. So far, there have been no reports on controlled drug release technology based on the motion of motile bacteria.

In this communication, as proof-of-concept, we will design and prepare an intelligent drug release system (I-DRS) that releases antimicrobial agents in the presence of motion of swarming P. mirabilis, and demonstrate the I-DRS efficacy against the spread of the swarmer colony. The I-DRS comprises of a mesoporous silica particle MCM-41 that contains an antimicrobial drug (Fig. 1). In the absence of motile bacteria, the therapeutic payload is retained within the pores and prevented from leaking out via a polymeric shell, electrostatically bound to the particle surface, that blocks the pore entrances. The polymer used is a pNIPAAm-co-pAEMA copolymer, consisting of the monomers N-isopropylacrylamide (NIPAAm) and 2-aminoethyl methacrylate hydrochloride (AEMA), conjugated to peptide ligands with sequence YVLWKRKRKFCFI-NH$_2$ that bind to lipopolysaccharides (LPS) (**3** in Fig. 2). In this way, when swarmer P. mirabilis cells approach close to the I-DRS, polymeric chains initially found on the surface of the particle will attach themselves onto the fast moving cells because of the LPS-copolymer **3** binding and be pulled away (i.e., mechanically stripped off the particle due to the strong force from rapidly moving bacteria); hence allowing the encapsulated drug to be released into the bacterial colony (Fig. 1). We term this bacteria-triggered drug release mechanism as Motion-Induced Mechanical Stripping (MIMS).

## Results and Discussion

**Copolymer 3 seals in payloads**. The copolymer pNIPAAm-co-pAEMA is used because it is non-toxic to P. mirabilis at the concentrations used here, soluble in water and covenient to synthesize and funtionalize. In particular, pNIPAAm-based copolymers and pAEMA have previously been utilized for drug delivery to various cells with little cytotoxicity effects[18–21]. The peptide YVLWKRKRKFCFI-NH$_2$ binds relatively strongly to LPS, found on the outer membrane of Gram-negative bacteria, with an association constant of $K_a = 0.130$ μM$^{-1}$ (Supplementary Fig. 1)[22–24]. The binding of the peptide to LPS is comparable to the association between a YVLWKRKRFIFI peptide, a sequence close to the one used in this study, and LPS ($K_a = 0.22$ μM$^{-1}$)[22], and between other LPS binding peptides and LPS (~0.1–0.2 μM$^{-1}$)[25,26].

Copolymer **3** was prepared by atom-transfer radical polymerization of NIPAAm and AEMA with ethyl-2-bromopropionate as initiator and CuBr as catalyst to yield copolymer **1** with $M_n = 10171$ g mol$^{-1}$ and $M_w = 21985$ g mol$^{-1}$ (Fig. 2)[27]. A fraction of the amine groups in **1** (~20%) were converted to maleimides by reacting with 3-maleimidopropionic acid to give copolymer **2**. Peptide ligands YVLWKRKRKFCFI-NH$_2$ were subsequently attached to **2** via a stable carbon (maleimide)-sulfur (cysteine) bond to obtain the desired copolymer **3** with $M_n = 14,212$ g mol$^{-1}$ and $M_w = 43,033$ g mol$^{-1}$ (Fig. 2). The average number of NIPAAm, AEMA and peptide units per chain is $m \sim 77$, $n \sim 11$ and $p \sim 2$, respectively. Copolymer **3** is positively charged with a zeta potential $\zeta_{polymer} = 21.8 \pm 1.3$ mV (s.d., three independent experiments) due to the presence of the remaining amine groups. The lower critical solution temperature LCST of copolymer **3** is determined to be 53 °C (Supplementary Fig. 2 and Supplementary Note 1) which is

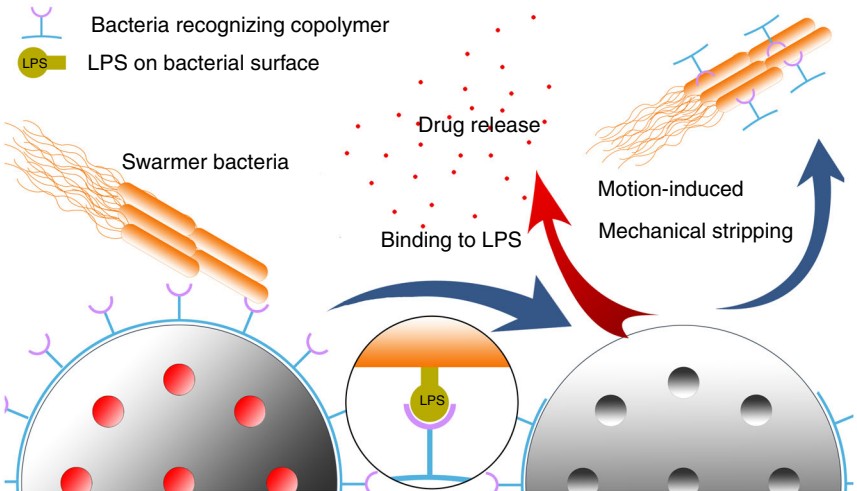

**Fig. 1** Mechanism of Motion-Induced Mechanical Stripping (MIMS). Antimicrobial agents are encapsulated within the pores of a mesoporous silica particle and prevented from leaking out by the bacteria recognizing copolymer attached onto the surface of the particle (left). When swarmer Gram-negative bacteria brush against the drug carrier, the polymeric chains bind selectively to the fast moving cells and are stripped off the surface of the particle; hence exposing the pores and allowing the payload to be released into the bacterial colony (right). Only the top hemisphere of the particle is shown

**Fig. 2** Synthetic steps for the preparation of copolymer **3**. From GPC and $^1$H NMR results, $m \sim 77$, $n \sim 11$ and $p \sim 2$

significantly higher than the temperature used in the experiments reported here (≤37 °C). The mesoporous silica particles (average diameter = 264 ± 60 nm), with an average mesopore diameter of 3.5 nm, were prepared using standard procedures (Supplementary Fig. 3)[28].

In order to show that copolymer **3** is able to seal in payloads, toluidine blue (TB) was first loaded into mesoporous silica particles, and the copolymer was electrostatically bound to the negatively charged particle surface (i.e., measured zeta-potential of MCM-41 $\zeta_{MCM-41} = -41.0 \pm 0.9$ mV (three independent experiments)) to form a copolymer shell. TB is a positively charged dye that can be readily quantified using spectroscopic methods such as UV–vis absorbance. The strong absorbance of TB at visible wavelengths makes it a convenient indicator to quickly assess the loading ability of the dye into polymer-coated silica particles. The dye-loaded particles were dialyzed for 4 days before treatment with 1 M HCl (5 min) to remove copolymer **3** chains from particle surface. A similar experiment was also conducted on dye-loaded silica particles but without an initial copolymer shell. For the latter, the absorption spectrum of TB released during the HCl treatment has low absorbance intensity (Fig. 3a); indicating that most of the dye was released after 4 days of dialysis and only a small amount retained within the particles. On the other hand, for dye-loaded particles with an initial copolymer shell, a significant amount of TB was released upon detachment of copolymer chains from the particle surface during the HCl treatment (Fig. 3a). In particular, for the 635 nm

absorption peak, the absorbance of TB released from particles with an initial copolymer shell is ~18 times larger than that in the absence of an initial shell. This clearly suggests that during the dialysis period, copolymer **3** is able to prevent leakages of TB stored within the pores of the silica particles. We therefore propose that copolymer **3** is effective for sealing in active compounds loaded into mesoporous silica particles.

In this study, the drug used is tobramycin which is an antibiotic known to be active against *P. mirabilis*[29,30]. From UV–vis absorption spectroscopy, the release curve of tobramycin from silica particles without a copolymer shell after the drug has been loaded into the particles is given in Fig. 3b. By fitting the release curve to a double-exponential rise function, the maximum amount of tobramycin released from the particles is estimated to be ~0.2 mg of drug per mg of particles which is comparable to the loaded amount of Tob in mesoporous silica particles (i.e., 0.234 mg of drug per mg of particles). We note that the antibiotic released from silica particles follows a two-step process; a fast release within the first 2 h followed by a slower release. According to the Higuchi model[31], the linear relationship observed between the amounts of drug released after time $t$ ($Q$) vs. $t^{1/2}$ in both steps (Supplementary Fig. 4 and Supplementary Note 2) suggests a diffusion controlled kinetics for the release of tobramycin. In the case of copolymer **3**-coated silica particles containing the drug, no detectable amount of tobramycin was released after dialysis (4 days); indicating that copolymer **3** prevented leakages of tobramycin from the particles.

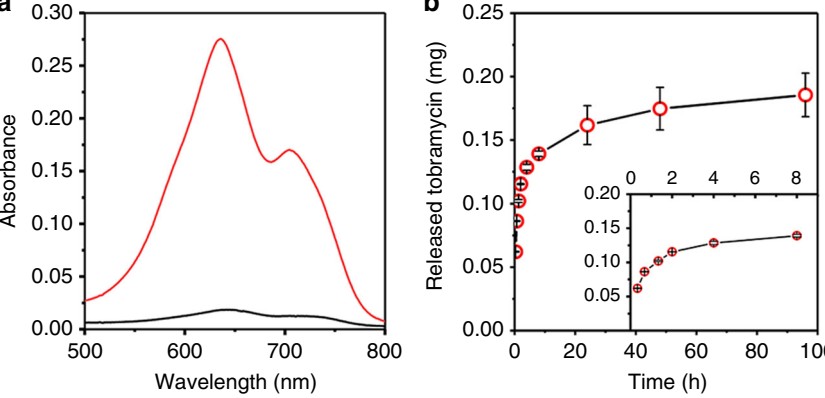

**Fig. 3** Loading and release of toluidine blue (TB) and tobramycin. **a** Absorption spectra of TB released after 4 days of dialysis and 1 M HCl treatment for mesoporous silica particles with (red) and without (black) an initial copolymer shell. **b** Drug release curve showing the amount of tobramycin released from drug-loaded mesoporous silica particles without coploymer shell at various times. The release curve for the first 8 h is given in the inset. Each data point is the average of two independent experiments and error bars represent SD

**MIMS in aqueous solution**. MIMS is first demonstrated in water using an LPS-coated polystyrene (PS) bead. In this case, PS latex beads (~1 μm mean particle size) were first treated with LPS to allow the adsorption of LPS onto the surface of the beads. When an aqueous mixture of drug-loaded copolymer **3**-coated silica particles (10 mg) and free LPS (2 mg mL$^{-1}$) was stirred at 100 rpm for 2 h, no released drug was detected via HPLC. On the other hand, when a mixture of drug-loaded copolymer **3**-coated silica particles and LPS-coated PS beads (10 mg) was stirred at 100 rpm to enable the two entities to come into contact with each other, tobramycin was released and the drug release kinetics is given in Supplementary Fig. 5. We note that ca. 0.15 mg of drug per mg of particles was released after stirring with LPS-coated PS latex beads which is in agreement with the amount of drug released from silica particles without an initial copolymer shell (Fig. 3b). It is also observed that in the absence of stirring, no released drug was detected in still water.

We propose that LPS on PS beads binds to copolymer **3** on silica particles when the two entities are in close contact. The copolymer is subsequently pulled off the surface of silica particles when the particles and PS beads are forced apart during stirring. The release of tobramycin occurs at a faster rate (>80% of steady-state amount released within 2 h) compared to the release of drug in Fig. 3b possibly due to more collisions occurring between the exposed silica particles and PS beads. Free LPS by itself is unable to cause detachment of copolymer **3** from silica particles due to insufficient stripping energy. When a mixture of amine-modified PS beads (~1 μm, 10 mg) that are *not* coated with LPS and drug-loaded copolymer **3**-coated silica particles was stirred at 100 rpm, no significant amount of released drug was detected even after 2 h; suggesting that in the absence of LPS on PS beads, MIMS is not in operation. Furthermore, collision between PS beads and silica particles alone does not lead to the detachment of copolymer chains from the latter.

**MIMS on agar surface**. The efficacy of the I-DRS against the expansion of swarmer *P. mirabilis* is studied next. Drug-free copolymer **3**-coated silica particles (Fig. 4a) and copolymer **3**-coated particles containing tobramycin (Fig. 4b, c) were separately deposited on the left side of the (1%) agar surface. The cells were subsequently inoculated on the right side of the agar and cultured for 20 h at 37 °C. An expanding swarmer colony was achieved after a lag time of ~8 h. When 0.8 mg of drug-free copolymer **3**-coated silica particles were used, the swarmer cells

colonized the entire agar surface (Fig. 4a). This suggests that in the absence of drug, the I-DRS is non-toxic when the amount used is as high as 0.8 mg.

When copolymer **3**-coated silica particles loaded with tobramycin (ca. 0.2 mg of drug per mg of particles) were used, the swarmer colony spread throughout the particle-free surface (i.e., right side) of the agar (Fig. 4b, c). However, an impediment to the expansion of the colony across the surface of the agar containing the I-DRS (i.e., left side) was observed. In addition, the swarmer colony was found to possess a secondary front (thin layer) that preceded a primary front (thick layer). Two parameters are employed to characterize the I-DRS efficacy: $L_s$ and $L_p$ are the distances from the center of the vertical line dividing the particle (left side) and particle-free (right side) zones to the intersection points between the horizontal bisector and the secondary and primary fronts, respectively. When 0.2 (Fig. 4b) and 0.05 (Fig. 4c) mg of silica particles were used, the secondary fronts of both colonies expanded only slightly into the particle zone with $L_s =$ 3.9 and 13.5 mm, respectively. On the other hand, both the primary fronts were not able to cross into the particle zone; $L_p =$ 7.1 and 2.3 mm for 0.2 and 0.05 mg of particles, respectively. Therefore, when swarmer cells from the secondary front enter into the particle zone and come in contact with the particles, drug is released. Bacteria in the secondary front are subsequently damaged and the colony is not able to continue its expansion. In the presence of a larger amount of particles, more tobramycin is released into the colony resulting in a greater impediment to colony expansion (i.e., shorter $L_s$ and longer $L_p$ for 0.2 mg particles compared to 0.05 mg particles). It is worth mentioning that tobramycin released from the silica particles deposited on the left side of the agar penetrates through agar, however, a concentration gradient is expected such that the amount of free tobramycin that has diffused away to the right-side of the agar is likely to be lower than the minimum inhibitory concentration (MIC) against *P. mirabilis* (~2 μg mL$^{-1}$) (Supplementary Fig. 6 and Supplementary Note 3), and does not cause inhibition of the proliferation of cells on that side of the agar.

A control experiment was also performed to examine the effects of immobile cells on the I-DRS. In this case, *P. mirabilis* cells were inoculated on the entire surface of a hard (4%) agar containing, on the left side surface, 0.2 mg of copolymer **3**-coated silica particles loaded with tobramycin (ca. 0.2 mg of drug per mg of particles). After the cells were dried and cultured for 20 h (at 37 °C), it was seen that *P. mirabilis* proliferated into a thick layer of immobile (vegetative) cells throughout the agar with no

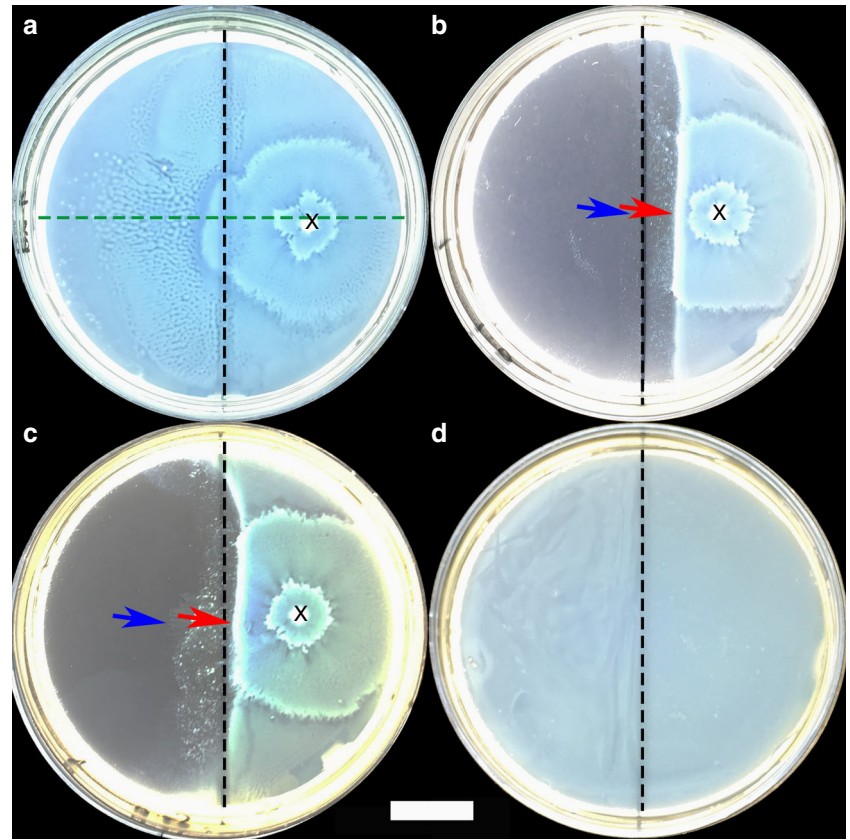

**Fig. 4** Release of drug via MIMS. Colony expansion (**a**–**c**) and proliferation (**d**) of *P. mirabilis* observed 20 h after cells were inoculated on 1% (**a**–**c**) and 4% (**d**) agar. Copolymer **3**-coated silica particles were deposited evenly on the left side of the agar surface. 0.8 mg of drug-free particles (**a**), and 0.2 (**b**, **d**) and 0.05 (**c**) mg of particles containing tobramycin were used. **a**–**c** *P. mirabilis* cells were inoculated on the right side of agar at point x. The vertical dotted line (black) is the boundary between the particle and particle-free zones that divides the agar into two equal halves, and the horizontal dotted line (green) in **a** bisects it. The blue and red arrows in **b** and **c** indicate the intersection points between the horizontal bisector with the secondary and primary colony fronts, respectively. **d** *P. mirabilis* were inoculated on the whole agar and immobile *P. mirabilis* cells were seen to proliferate throughout the agar. Scale bar: 2 cm

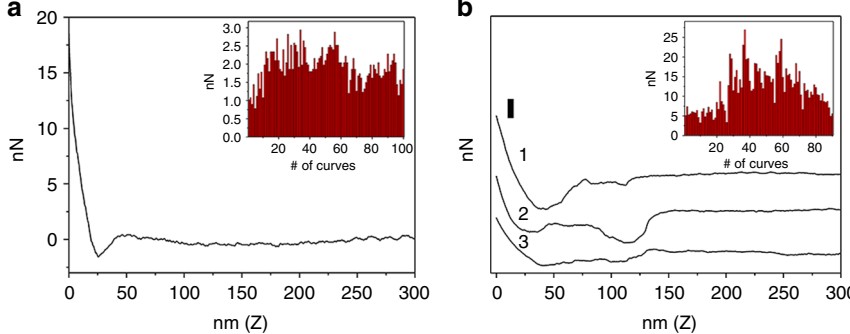

**Fig. 5** AFM force spectroscopy. **a** A representative force curve plotted for the interaction between a bare SiO$_2$ spherical tip and a layer of copolymer **3** attached on a Si wafer. The adhesion force histogram for 100 force curves is given in the inset. **b** Representative force curves plotted for the interaction between a copolymer **3** functionalized SiO$_2$ tip and a layer of LPS attached on a Si wafer and collected before 90 ramp cycles (adhesion forces for curves 1, 2 and 3 are 17.07, 16.36 and 7.10 nN, respectively). The adhesion force histogram before 90 ramp cycles is given in the inset. Scale bar: 10 nN

obvious inhibition of cell growth at the side of the surface containing the particles (Fig. 4d). When a similar experiment was performed using drug-loaded silica particles *without* a copolymer shell, the leakage of tobramycin into the surroundings prevented the proliferation of bacteria in the particle-zone (Supplementary Fig. 7 and Supplementary Note 4). Since differences in agar concentration do not significantly affect the inhibition zones and antibiotic gradient[32], the absence of bacterial inhibition zones in

Fig. 4d indicates that tobramycin is not released by immobile bacteria and this is in agreement with our earlier findings that when copolymer chains are not stripped off the particle surface, minimal leakage of payload occurs (see Fig. 3a).

Despite the antimicrobial effects of YVLWKRKRKFCFI-NH$_2$ against *P. mirabilis* (MIC = 1.6 mg mL$^{-1}$, Supplementary Fig. 8 and Supplementary Note 5), Fig. 4a shows that the peptide from the I-DRS does not cause obvious damage to the swarmer colony;

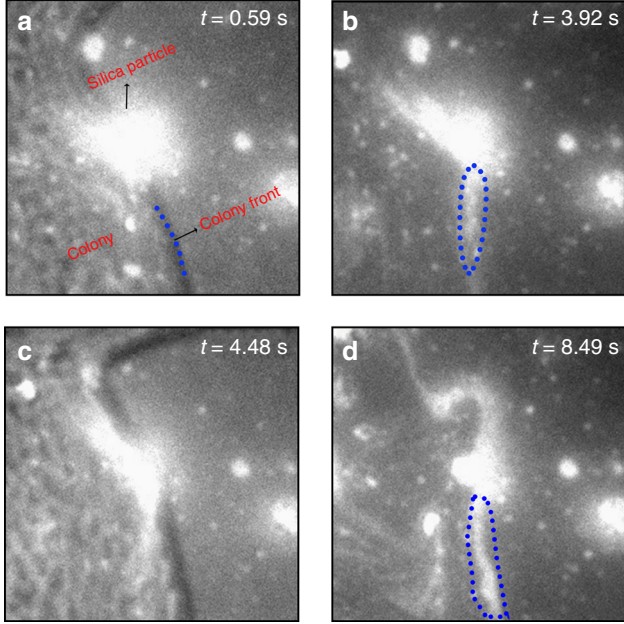

**Fig. 6** Visualizing MIMS in real-time. Wide-field fluorescence microscopy images of a dye-labelled copolymer **3**-coated silica particle at ($x = 50\,\mu m$, $y = 83\,\mu m$) in Supplementary Movie 1. **a**, **c** Show the overlap of the swarmer colony and labelled copolymer **3** recorded with white and red light irradiation, respectively, and **b**, **d** show only the images of copolymer **3**. Examples of fluorescent streaks arising from removed copolymer **3** are enclosed within blue dotted lines in (**b**) and (**d**). The time of each image $t$ is given. The image contrast is enhanced resulting in particles appearing larger than their actual size due to light scattering

probably because of its low concentration (<MIC). Furthermore, using an E-test assay, it is demonstrated that the MIC of tobramycin against vegetative *P. mirabilis* ($\sim 2.0\,\mu g\,mL^{-1}$) remained unchanged in the presence of a comparable amount of copolymer **3** as utilized in Fig. 4b (Supplementary Fig. 6 and Supplementary Note 3). Therefore, the bacteria do not become more susceptible nor the activity of the antibiotic enhanced in the presence of the copolymer used.

**Copolymer 3 adheres stronger to LPS than to silica.** AFM force spectroscopy experiments were conducted to determine the adhesion forces between copolymer **3** and $SiO_2$ and LPS. It is observed that the interaction between a bare $SiO_2$ spherical tip and a layer of LPS attached on the surface of a Si wafer is negligible (see Supplementary Fig. 9 for a representative force curve and Supplementary Note 6). On the other hand, significant interaction between the bare $SiO_2$ tip and a layer of copolymer **3** on a Si wafer is observed, where a representative force curve in Fig. 5a yields an adhesion force of 1.26 nN. A histogram of the adhesion force between the bare $SiO_2$ tip and copolymer **3**, based on 100 force curves (inset of Fig. 5a), shows that the adhesion force ranges from 0.78 to 2.94 nN.

In a separate experiment, a $SiO_2$ spherical tip functionalized with a shell of copolymer **3** was used to determine the adhesion force between the copolymer and LPS. We observed a relatively large adhesion force between the modified $SiO_2$ tip and LPS as illustrated by three representative force curves collected before 90 ramp cycles in Fig. 5b. The adhesion peaks are broad due to the stretching of outer copolymer upon interaction with LPS on the Si wafer. A histogram of adhesion force, based on 90 force curves, indicates that the adhesion force between copolymer **3** and LPS ranges between 3.26 and 26.92 nN (inset of Fig. 5b). For the 90 to

432 ramp cycles, the adhesion peaks for LPS become narrower and a much smaller force is seen (e.g., 1.7 to 3.5 nN for three representative curves in Supplementary Fig. 10 and Supplementary Note 6). After 432 ramp cycles, no adhesion peak is observed (Supplementary Fig. 10 and Supplementary Note 6). This is likely due to the peeling off of copolymer **3** from the contact surface of the $SiO_2$ tip as a result of the stronger adhesion force between the copolymer and LPS. We, therefore, propose that copolymer **3**, initially adsorbed on the surface of silica particles, will adhere more strongly to swarmer *P. mirabilis* in the secondary front when the latter comes into contact with the silica particles. The kinetic energy from the motile bacteria is able to strip the copolymer off the surface of the silica particles (i.e., MIMS is in operation) as demonstrated experimentally in the wide-field fluorescence microscopy (WFFM) measurements below.

**Real-time monitoring of MIMS using WFFM.** In the WFFM study, copolymer **3**, labelled with a photostable Atto 647N chromophore (ATTO-TEC GmbH), is electrostatically bound to drug-free particles. Supplementary Movie 1 shows a typical swarmer *P. mirabilis* colony expanding from left to right on an (1%) agar containing copolymer **3**-coated silica particles. Copolymer **3** chains were detached from silica particles found inside the swarmer colony (i.e., white fluorescent streaks flowing away from particles and swirling inside the colony) and not from particles located outside the colony. An example is given in Fig. 6 for the particle at ($x = 50\,\mu m$, $y = 83\,\mu m$). An overlay of the images of the swarmer colony and labeled copolymer at time $t = 0.59\,s$ is seen in Fig. 6a where the colony front is in contact with the particle. As the colony continues to expand, copolymer **3** is stripped off the particle surface by the swarmer cells (i.e., at $t = 3.92\,s$ where only the silica particle is imaged with no image interference from the colony (Fig. 6b) and at $t = 4.48\,s$ where images of both the colony and particle are recorded (Fig. 6c)). As time progresses, more copolymer chains are removed and swirls of fluorescent white streaks arising from the transportation of removed copolymers by the motile cells are visible (Fig. 6d). When the cells are immotile (Fig. 4d), MIMS is not in operation which prevents both the release of drug and inhibition of cell proliferation.

**Swarmer B. subtilis colony.** The MIC value of tobramycin against vegetative Gram-positive *Bacillus subtilis* is $\sim 1.5\,\mu g\,mL^{-1}$ (Supplementary Fig. 11 and Supplementary Note 7) which is slightly lower than the MIC against *P. mirabilis*. An experiment similar to Fig. 4c was therefore conducted on swarmer *B. subtilis* to compare its effects on the I-DRS. In this case, the colony is capable of expanding fully into the left side of an (0.5%) agar containing 0.05 mg of drug-loaded copolymer **3**-coated silica particles on the agar surface (Supplementary Fig. 11 and Supplementary Note 7); suggesting that no lethal amounts of drug are released from the particles despite the presence of motile *B. subtilis*.

Weaker binding is observed between the peptide YVLWKRKRKFCFI-NH$_2$ and lipoteichoic acid (LTA) found on the cell walls of *B. subtilis* ($K_a = 0.042\,\mu M^{-1}$, Supplementary Fig. 12). In addition, a surface charge neutralization by zeta potential study was performed. In the absence of copolymer **3**, *P. mirabilis* exhibits a larger negative potential ($\sim -20\,mV$) as compared to *B. subtilis* ($\sim -4\,mV$). As shown in Fig. 7, the addition of ca. 1–2 $\mu M$ of copolymer **3** neutralizes the negative charge of the bacterial cell wall. A larger over compensation of surface charges is seen for *P. mirabilis* (e.g., zeta potentials of +7 mV and +2 mV for *P. mirabilis* and *B. subtilis*, respectively, are obtained when the concentration of copolymer **3** added is 8 $\mu M$).

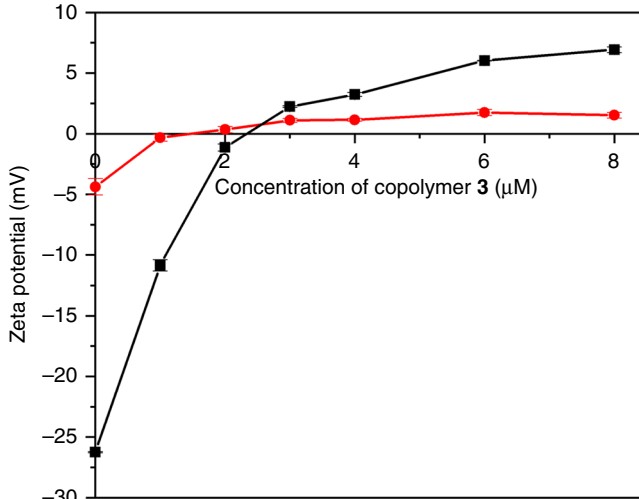

**Fig. 7** Surface charge neutralization by zeta potential study. Changes of zeta potential of *P. mirabilis* (black box) and *B. subtilis* (red circles) cells membrane in the presence of copolymer **3**. Each data point is the average of two independent experiments and error bars represent SD

This means that for a given concentration of copolymer **3** added, the number of copolymer **3** bound to *P. mirabilis* is significantly higher than that for *B. subtilis*.

The AFM force curves for interactions between bare SiO$_2$ spherical tip and LTA, and between LTA and copolymer **3** adsorbed on a SiO$_2$ spherical tip do not display adhesion peaks (Supplementary Figure 9 and Supplementary Note 6); suggesting that the adhesion force between copolymer **3** and LTA is too weak to be measured by AFM. We, therefore, propose that when swarmer *B. subtilis* come into contact with the copolymer on silica particles, the copolymer remains attached onto the particles. Since the copolymer is not adhered to motile *B. subtilis*, the motile cells are unable to strip the copolymer off the silica particle surface. This mechanism is in line with the WFFM study that shows a distinct absence of bright streaks moving away from drug-free silica particles coated with dye-labelled copolymer **3** within a swarmer *B. subtilis* colony (Supplementary Movie 2 for multi-layer colony and 3 for single-layer colony)[33,34]. This demonstrates two points: (i) collision by fast moving bacteria is not sufficient to cause the electrostatically bound copolymers to be detached from the silica particle surface, and (ii) MIMS is not in operation here.

In summary, the I-DRS proposed in this study releases therapeutic drugs upon sensing the presence of swarmer *P. mirabilis* and is able to effectively impede the spread of the colony. An important feature of the I-DRS is its affinity to *P. mirabilis* due to the relatively strong binding between the conjugated copolymer-(YVLWKRKRKFCFI-NH$_2$) system with the Gram-negative bacteria. Further studies are underway to realize the applications of MIMS in biomedicine where the short and long term toxicological effects of the I-DRS in humans are noted[35,36].

## Methods

**Chemicals and instruments**. Cetyltrimethylammonium bromide, AEMA, β-ala-nine, NIPAAm, ethyl-2-bromopropionate, maleic anhydride, *N,N,N',N',N'*-penta-methyldiethylenetriamine (PMDETA), tobramycin, TB, CuBr, tetraethoxysilane, lipoteichoic acid (LTA from *B. subtilis*, protein content ≤ 3%), lipopolysaccharide (LPS from *P. mirabilis*, protein content ≤ 3%, and *P. aeruginosa*), PS latex beads and amine-modified PS beads were purchased from Sigma-Aldrich. ATTO-647N-NHS ester was purchased from ATTO-TEC GmbH. Twenty-five percent ammonia was purchased from Sinopharm Chemical Reagent Co. Ltd. *o*-Benzotriazol-1-yl-*N,N,N',N'*-tetramethyluronium hexafluorophosphate (HBTU) and amino acids used

for solid phase peptide synthesis were purchased from GL Biochem (Shanghai) Ltd. NIPAAm was recrystalized using hexane before use. Dicarboxy terminated poly(N-isopropyl acrylamide) (P9110E-NIPAM2COOH, Mn: 10,000) was purchased from polymer source. All other chemicals were used as received. Tobramycin E-test strips were obtained from bioMérieux.

Transmission electron microscopy (TEM) images of silica particles were recorded using JEM 1400 (JEOL). Mass spectra were measured using a LCQ Fleet mass spectrometer (Thermo Scientific). Nuclear magnetic resonance (NMR) spectroscopy was carried out on an Ultrashield 400 Plus NMR spectrometer (Bruker). Gel permeation chromatography (GPC) data was obtained using a 1260 Infinity II (Agilent Technologies) GPC instrument with PolarGelM column (Agilent Technologies). UV–Vis absorption spectroscopy was measured using a Cary-100 UV–Vis spectrometer (Varian). Zeta potential was measured using a Zetasizer NANO system (Malvern). High performance liquid chromatography (HPLC, Waters 2695 Separations Module) was used to quantify the amounts of tobramycin released from copolymer **3**-coated silica particles.

**Synthesis of mesoporous silica MCM-41 particles**. In a 250 mL flask, a mixture of cetyltrimethylammonium bromide (0.35 g) and 25% ammonia (40 mL) were added into water (70 mL) and stirred at 80 ℃ for 30 min[28]. Tetraethoxysilane (1.70 g) was then slowly added to the mixture and stirred at 80 °C for 2 h. The white precipitate was filtered and washed with water (four times) and ethanol (four times). The mesoporous silica particle MCM-41 product was dried overnight at 80 °C and calcinated at 600 ℃ for 6 h at a heating rate of 1 °C min$^{-1}$.

**Synthesis of bacterial recognizing peptide**. The peptide sequence YVLWKRKRKFCFI-NH$_2$ was synthesized using solid phase synthesis method[24]. MS (TOF) found: 1786.99; calculated: 1787.03 for C$_{88}$H$_{136}$N$_{23}$O$_{15}$S [YVLWKRKRKFCFI-NH$_2$+H]$^+$.

**Synthesis of 3-maleimidopropionic acid 4**. In a 250 mL flask, β-alanine (0.89 g) and maleic anhydride (0.98 g) were mixed in dry THF (150 mL) and stirred for 2 days. The solvent was evaporated and toluene (100 mL) was used to suspend the white precipitate. The suspension was then stirred at 120 °C for 6 h and cooled to room temperature. Toluene was then removed by evaporation to leave a yellow precipitate. The product 3-maleimidopropionic acid **4** was first extracted from the yellow precipitate using DCM. The DCM solution was washed with NaHCO$_3$ solution (three times) followed by water and brine. The organic phase was dried by Na$_2$SO$_4$ and evaporated to get the crude product which was purified using flash column with DCM. See Supplementary Fig. 13. $^1$H NMR (CDCl$_3$): δ(ppm) 6.72(s, 2H), 3.84 (t, 2H), 2.71 (t, 2H). $^{13}$C NMR(CDCl3): δ(ppm) 176.2, 170.3, 134.3, 33.3, 32.5.

**Synthesis of copolymer 1**. A mixture of NIPAAm (1.98 g, NIPAAm), 2-aminoethyl methacrylate hydrochloride (0.41 g, AEMA), ethyl-2-bromopropionate (0.018 g), *N,N,N',N',N'*- pentamethyldiethylenetriamine (45 μL, PMDETA), methanol (16 mL) and water (24 mL) in a 100 mL schlenk flask was degassed by cycles of freeze-pump-thaw[27]. CuBr (0.015 g) was added while the mixture was frozen. The solution was then stirred under argon atmosphere at room temperature for overnight. The product was purified by dialysis and dried to obtain pNIPAAm-*co*-pAEMA copolymer **1**. See Supplementary Fig. 14. The number average mole-cular weight $M_n = 10171$ g mol$^{-1}$, weight average molecular weight $M_w = 21,985$ g mol$^{-1}$ and polydispersity PD = 2.16.

**Synthesis of copolymer 2**. O-Benzotriazol-1-yl-N,N,N',N'-tetramethyluronium hexafluorophosphate (0.02 g, HBTU) and 3-maleimidopropionic acid **4** (0.085 g) were dissolved in dry DMF (5 mL) and stirred for 10 min before being added to a solution containing copolymer **1** (0.24 g) in dry DMF (10 mL). The reaction mixture was stirred for 2 days at room temperature. DMF was then evaporated and the crude product was dissolved in water and purified by dialysis. The solution was lyophilized to obtain dry copolymer **2**. See Supplementary Fig. 15.

**Synthesis of copolymer 3**. Peptide YVLWKRKRKFCFI-NH$_2$ (100 mg) was dis-solved in PBS buffer (100 mL) and filtered using a 25 mm syringe filter with 0.45 μm membrane. Copolymer **2** (0.2 g) was then added into the filtered solution. The mixture was stirred for 3 days at room temperature and purified by dialysis. The final solution was lyophilized to yield copolymer **3**. See Supplementary Fig. 16. The number average molecular weight $M_n = 14,212$ g mol$^{-1}$, weight average molecular weight $M_w = 43,033$ g mol$^{-1}$ and polydispersity PD = 3.63. Assuming a stoichiometric reaction, the ratio *m:n* = 7.06:1. This agrees with the $^1$H NMR data of copolymer **3** in D$_2$O where peaks at δ(3.90 ppm) from $H_a$ and δ(4.25-4.10 ppm) from $H_b$ and $H_c$ have an integration area ratio = 5.74:1. The values of *m*, *n*, and *p* were therefore taken to be ca. 77, 11 and 2, respectively.

**Fluorescent dye labeling of copolymer**. Copolymer **2** (200 mg) was dissolved in 0.1 M NaHCO$_3$ solution (15 mL). ATTO-647N-NHS ester (1 mg) was added into the solution and stirred for 2 days at room temperature. The mixture was then dialyzed and lyophilized to get copolymer **5**. A similar procedure as outlined above

was performed on copolymer **5** to yield the dye-labelled copolymer **6** (Supplementary Fig. 17). On average, one dye molecule was conjugated to a copolymer chain.

### LCST of copolymer 3

Transmittance at 450 nm of copolymer **3** dissolved in PBS buffer (~0.33 g per mL) with different pH values (2.7, 5.8, and 7.2) was measured at different temperatures using a Cary-300 UV–Vis spectrometer (Varian) equipped with a dual cell peltier accessory. The lower critical solution temperature LCST is the temperature at 50% light transmittance at 450 nm of the original polymer solution. The LCST of commercial PNIPAAm ($\alpha,\omega$-bis(carboxy)-terminated) was also measured as a control.

### Determining the loaded amount of drug

A weighed amount of silica particles (~5 mg) were loaded with tobramycin by stirring an aqueous solution (~500 μL) of tobramycin (400 mg mL$^{-1}$) with the particles for 3 days. The drug-loaded particles were recovered by centrifugation (10,956 × $g$), washed with DI water (0.5 mL), freeze dried and weighed again. The loading amount was obtained by the difference in mass of silica particles before and after loading with tobramycin. For our experiment, the amount of tobramycin loaded is 0.234 ± 0.006 mg of drug per 1 mg of silica particles (average of two independent experiments).

### E-test assay

For *P. mirabilis*: An (4%) agar plate was prepared in a 10-cm Petri dish. An exponential-growth phase *P. mirabilis* culture was applied on the entire agar surface and dried for 15 min. A tobramycin *E*-test strip was placed on the center of the surface of the agar before incubation for 20 h at 37 °C.

To examine if copolymer **3** affects the viability of *P. mirabilis* when exposed to tobramycin, 1.4 mg mL$^{-1}$ copolymer **3** (200 μL) was evenly applied on the entire surface of an (4%) agar and dried for 30 min. An exponential-growth phase *P. mirabilis* culture was then applied on the entire agar surface and dried for 15 min. A tobramycin *E*-test strip was placed on the center of the agar followed by an incubation period of 20 h at 37 °C. The MIC value was recorded following manufacturer instructions.

For *B. subtilis*: An (4 %) agar plate was prepared in a 10-cm Petri dish. An exponential-growth phase *B. subtilis* culture was applied on the entire agar surface and dried for 15 min. A tobramycin *E*-test strip was placed on the center of the surface of the agar before incubation for 20 h at 37 °C. The MIC value was recorded following manufacturer instructions.

### MIC of YVLWKRKRKFCFI-NH₂ against P. mirabilis

The MIC of the peptide YVLWKRKRKFCFI-NH$_2$ against *P. mirabilis* was determined. Mid logarithmic phase of an overnight culture in LB broth was prepared and OD$_{600}$ adjusted to 0.2. The bacterial medium (50 μL) was added to an equal volume (50 μL) of an aqueous solution of peptide of various concentrations (i.e., 0, 64, 128, 256, 320, 640, 1280, 1600, 1920, and 3200 μg mL$^{-1}$), The mixture was incubated for 3 h at 37 °C, and a 100 times-diluted solution (10 μL) was applied onto an LB-agar plate. The minimum concentration of peptide that results in 90% growth inhibition is taken to be the MIC value.

### Isothermal titration calorimetry

The binding constants between peptide YVLWKRKRKFC-NH$_2$ and LPS and between the peptide with LTA were determined using a MicroCal iTC200 isothermal titration calorimeter. Peptide, LPS (from *P. mirabilis*) and LTA (from *B. subtilis*) solutions were first prepared by dissolving each component in 10 mM phosphate buffer (pH 7.0). LTA (50 μM) was loaded into the sample cell, the reference cell was filled with the same buffer and the syringe was filled with 1 mM peptide solution. Typically, 25 injections of 1.6 μL of peptide were made into the sample cell at 25 °C and the sample cell was stirred at 900 rpm. Raw data were obtained and analyzed using single set of binding sites in the MicroCal PEAQ-ITC analysis software. Association constant ($K_a$) was directly determined from the ITC profiles.

### Dye and drug release experiments

The capability of copolymer **3** to encapsulate a payload within mesoporous silica particles was demonstrated by performing UV–vis absorption spectroscopy on TB dye retained in the particles. TB was loaded into the silica particles by dissolving the dye (1 mg) in water (1 mL) followed by the addition of silica particles (50 mg). Following constant stirring at room temperature for 2 days, the suspension (100 μL) was added into an aqueous solution (1 mL) of copolymer **3** (20 mg mL$^{-1}$) to allow the latter to electrostatically adsorb onto the particle surface. A control experiment was also conducted by adding the suspension (100 μL) into water (1 mL) in the absence of copolymer **3**. Both samples were subsequently dialyzed for 4 days, centrifuged (5590 × $g$) and the precipitates were re-suspended in 1 M HCl (1 mL) for 5 min at room temperature. In the case of silica particles with a copolymer shell, the HCl treatment facilitates the detachment of the copolymer from the particle surface. The suspensions were subsequently centrifuged (5590 × $g$) and the supernatants containing released TB were diluted (five times) before running the UV–Vis absorption spectroscopy.

The drug release curve was constructed by measuring the amounts of tobramycin released from drug-loaded mesoporous silica particles *without* polymeric shell at various times. Silica particles were loaded with tobramycin by

stirring an aqueous solution (100 μL) of tobramycin (400 mg mL$^{-1}$) with silica particles (1 mg) for 3 days. The particles were centrifuged (10,956 × $g$), washed with DI water and re-suspended in DI water (1 mL) with continuous shaking at 37 °C. At each measurement time point (i.e., 20 min, 40 min, 80 min, 2 h, 4 h, 8 h, 24 h, 48 h and 96 h), the absorption spectrum of the supernatant after centrifugation (10,956 × $g$) was measured using a UV–Vis absorption spectrometer. The absorbance of tobramycin at 210 nm was used to determine the concentration of the drug[37]. The cumulative amount of tobramycin released at a particular measurement time was determined by summing the amount measured at that time and the cumulative amount measured at the last measurement time. After each absorption measurement, the silica particles, recovered by centrifugation (10,956 × $g$), were re-suspended in DI water (1 mL) and shaken. At the next measurement time, the absorption spectrum of the supernatant was again recorded. The total amount of tobramycin released from 1 mg of silica particles was estimated by fitting a double-exponential rise function to the cumulative release curve (Fig. 1b) (i.e., $y = -(A_1\exp(-k_1t) + A_2\exp(-k_2t)) + C$, where $C$ is the total amount of drug released).

A similar experiment was conducted for copolymer **3**-coated silica particles loaded with drug. Since the copolymer absorbs in the same region as tobramycin, a HPLC instrument (Waters 2695 Separation Module) equipped with a PDA detector set at 210 nm (Waters 2996 PDA) was used instead to determine the presence of released drug. An isocratic mobile phase consisting of a 0.05 M sodium hydrogen phosphate buffer (pH = 10) was used. Chromatography was carried out at 25 °C on an Alltima™ C 18 5 u 250 mm × 4.6 mm column and the compounds eluted at a steady flow of 0.75 mL min$^{-1}$. The retention time of tobramycin is ~61 min. For copolymer **3**-coated silica particles loaded with drug, no tobramycin peaks were observed after dialysis (4 days).

### Drug release with LPS-coated PS beads

A HPLC experiment was conducted to measure the amounts of tobramycin released from copolymer **3**-coated mesoporous silica particles loaded with drug when stirred together with LPS-coated PS beads in water. The HPLC calibration curve was first constructed by measuring the peak area of tobramycin (peak at 60–70 min retention time) at various concentrations (i.e., 0.5, 1.0, 1.5, and 2.0 mg mL$^{-1}$). The LPS-coated PS beads were prepared following the procedure in ref. [38] Basically, 10 mg PS latex beads (~1 μm mean particle size) were suspended in an aqueous solution (1 mL) of LPS (2.0 mg mL$^{-1}$) for 20 min. The drug-loaded copolymer **3**-coated silica particles (containing 10 mg of silica particles) were subsequently added into the LPS-coated PS solution and continuously stirred at 100 rpm at room temperature. At each measurement time point (i.e., 5 min, 10 min, 15 min, 20 min, 40 min, 80 min, 2 h and 4 h), the HPLC peak of the supernatant after centrifugation (10,956 × $g$) was measured. The amount of released tobramycin was determined using the calibration curve. Three control experiments were conducted: (i) an aqueous mixture (1 mL) containing drug-loaded copolymer **3**-coated silica particles and free LPS (2.0 mg mL$^{-1}$) was stirred at 100 rpm for 2 h, (ii) an aqueous mixture (1 mL) containing drug-loaded copolymer **3**-coated silica particles and LPS-coated PS particles was allowed to be still for 2 h without stirring and (iii) an aqueous mixture (1 mL) containing drug-loaded copolymer **3**-coated silica particles and amine-modified PS latex beads (~1 μm mean particle size and without LPS coating) (containing 10 mg beads) was stirred at 100 rpm for 2 h.

### Colony expansion and proliferation tests

Exponential growth phase *P. mirabilis* (ATCC 7002) and *B. subtilis* (ATCC 6051) were used. The liquid culture media was lysogeny broth (LB). The exponential-growth phase was obtained by diluting an overnight culture 20 times and then cultured for a further 3 h. When an exponential-growth phase was obtained, the OD$_{600}$ of the culture was 0.9 and the bacteria number was determined to be 10$^7$ CFU mL$^{-1}$. The solid culture media was LB with 0.2% glucose and 1% or 4% agar for *P. mirabilis*, and LB with 0.5% agar for *B. subtilis*.

To demonstrate that copolymer **3**-coated silica particles in the *absence* of loaded tobramycin is non-toxic against swarmer *P. mirabilis*, LB-agar solid media (1%) was first prepared in 10-cm diameter Petri dishes. The surface of the agar was divided into two equal parts. 0.8 mg of drug-free copolymer **3**-coated silica particles were evenly applied onto the left side of the agar using an L-shape spreader and dried for 30 min before a fresh exponential-growth phase *P. mirabilis* culture (1 μL) was inoculated at a point on the right side of the agar (particle-free zone) (see Fig. 4a). The dish was dried for a further 30 min before incubation for 20 h at 37 °C.

To examine the effects of released tobramycin on swarmer *P. mirabilis*, the drug was first loaded and encapsulated within the silica particles. In this case, silica particles (1 mg) and tobramycin (400 mg mL$^{-1}$) were mixed in water (100 μL) and stirred for 3 days. The suspension was then centrifuged (894 × $g$) and the precipitate was re-suspended in water (100μL) containing copolymer **3** (25 mg mL$^{-1}$). The suspension was centrifuged (894 × $g$) and the precipitate was re-suspended in fresh DI water to achieve a particle concentration ranging from 1 to 0.25 mg mL$^{-1}$.

Solid LB-agar media (1%) was prepared in 10-cm diameter Petri dishes. The surface of the agar was divided into two equal halves. An aqueous suspension of copolymer **3**-coated particles (200 μL) was applied on the left half surface of the agar using an L-shape spreader to achieve various amounts of deposited particles (i.e., 0.2–0.05 mg of silica particles). The suspension was dried for 30 min before a fresh exponential-growth phase *P. mirabilis* culture (1 μL) was inoculated at a point

on the right side of the agar (particle-free zone) (see Fig. 4b, c). The dish was dried for a further 30 min before incubation for 20 h at 37 °C.

To demonstrate that the drug release is not triggered by immobile cells, hard (4%) LB-agar was used. To prevent swarming *P. mirabilis* from undergoing differentiation, a harder agar (e.g., 4%) can be used[5,32]. It is worthwhile to note that differences in agar concentration do not significantly affect the inhibition zones and antibiotic gradient of free tobramycin[32]. An exponential-growth phase *P. mirabilis* culture (800 μL) was evenly inoculated on the entire agar surface and dried for 4 h to form a layer of immobile bacteria (see Fig. 4d). Drug-loaded silica particles (0.2 mg) with and without a copolymer shell were deposited on the left side of the agar. The dish was then incubated for 20 h at 37 °C.

For the swarmer *B. subtilis* experiment, an aqueous suspension (200 μL) of copolymer **3**-coated silica particles loaded with tobramycin (0.05 mg) was first applied on the left half surface of a (0.5 %) LB-agar and dried for 30 min[33,34]. A fresh exponential-growth phase *B. subtilis* culture (1 μL) was inoculated at a point on the right side of the agar. The dish was dried for a further 30 min before incubation for 20 h at 37 °C. All experiments described here were repeated at least four times and the observations were found to be reproducible.

**AFM force spectroscopy.** Si wafers were cleaned by sonication in acetone solvent for 30 min, rinsed with DI water and dried by $N_2$ gas. A thin superglue liquid film was applied on the surface of the Si wafer by spin-coating. LPS (from *P. mirabilis*), LPA (from *B. subtilis*) and copolymer **3** were made into thick films by freeze drying before they are attached onto the thin superglue film. The samples were left to further dry overnight, and any samples unattached were washed away by rinsing thoroughly with PBS buffer solution.

$SiO_2$ AFM probes (spherical with diameter of 650 nm ± 10%) were purchased from TipsNano. A similar method as used for modifying silica particles was used to functionalize the $SiO_2$ AFM probe with a shell of copolymer **3**. The modified probe was washed with PBS buffer to remove any unattached copolymer. The spring constants of bare probe and modified probe were calibrated and values of 3.6 and 5.2 N m$^{-1}$, respectively, were calculated by the built-in thermal tune. The force curves were measured in PBS buffer solution using an AFM instrument (Bruker, MultiMode 8). The adhesion forces between bare $SiO_2$ probe and copolymer **3**/LPS/LTA attached onto the wafer were measured. In addition, the adhesion forces between copolymer **3** adsorbed on the $SiO_2$ probe and LPS/LTA attached onto the wafer were also measured.

**Surface charge neutralization by zeta potential study**. An overnight *P. mirabilis* culture (50 μL) was re-suspended in fresh LB media (5 mL) and continuously shaken at 180 rpm for 3 h at 37 °C. The culture was re-suspended in a phosphate buffer to afford $OD_{600}$ ~ 0.2. 700 μL of the bacterial culture was then used in the zeta potential experiment. Increasing amounts of copolymer **3** were subsequently added and the resulting zeta potentials measured. A similar procedure was also used to measure the zeta potentials of *B. subtilis* in the presence of varying amounts of copolymer **3**.

**Wide-field fluorescence microscopy setup.** The wide-field fluorescence microscopy consists of an inverted microscope (IX 71, Olympus) coupled to a highly sensitive CCD camera (CascadeII 512B, Photometrics). A white light source was fixed above the bacteria sample and used to observe bacteria motion and a 633 nm HeNe laser source (35 mW, Melles Griot) was used for fluorescence microscopy. The laser light was circular polarized and filtered by an excitation filter (Z633/10, Chroma) before being focused onto the back-focal plane of the objective lens. The transmitted light and fluorescence from the sample were passed through an air objective lens (20×, N.A. = 0.40, Olympus), a dichroic mirror (Z633rdc, Chroma), an emission filter (HQ645lp, Chroma) and a 3.3 × camera lens before entering the CCD camera. The dimension of an image frame was measured to be 120 × 120 μm$^2$ (512 × 512 pixels) using a stage micrometer and the rate of the CCD camera is 30 frames per second.

An aqueous solution (20 μL) of copolymer **3** (25 mg mL$^{-1}$) was mixed with an aqueous suspension (10 μL) of drug-free silica particles (25 mg mL$^{-1}$) and incubated at room temperature for 30 min before centrifugation (894 × g). The precipitate was washed using water and re-suspended in water (1 mL). A drop of the particle suspension (1 μL) was deposited on the surface of 1 or 0.5 % LB-agar and dried for 30 min. A freshly prepared *P. mirabilis* exponential-growth phase culture (1 μL) was inoculated close to the particle zone on the 1% agar. The agar was dried for 30 min before incubation at 37 °C for 10 h to form a swarmer colony. For swarmer *B. subtilis*, the cells were inoculated close to the particle zone on the 0.5% agar and dried for 30 min before incubation at 37 ºC for 3 h to form a swarmer colony. Wide-field fluorescence microscopy was used to visualize the interaction between dye-labelled copolymer chains on the particle surface and motile swarmer *P. mirabilis* and *B. subtilis*.

**Data availability**

The data that support this study is available from the corresponding author upon reasonable request.

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

## Acknowledgements

The authors thank the Nanyang Technological University for financial support. A MOE Tier 1 grant (RG6/15) is acknowledged.

## Author contributions

E.K.L.Y. designed and conceived the experiments with inputs from S.L. S.B. and B.X., and S.L., W.B., Q.D., S.S., X.W. and A.S. performed the experiments and analyzed the data. The paper was written by E.K.L.Y., and all authors discussed the results and commented on the manuscript.

## Additional information

**Competing interests:** The authors declare no competing interests.

