## [Peer Review File · Nature Communications]

Reviewers' comments:

Reviewer #1 (Remarks to the Author):

Manuscript Number: NCOMMS-18-14211

Title: Lipopolysaccharide-affinity copolymer senses the rapid motility of swarmer bacteria to trigger antimicrobial drug release

First, I would like to thank the Editor for the invitation to review.

The paper from Shengtao Lu et al. reports the drug release system based on pNIPAAm-co-AEMA copolymer that prevents drug leakage in an antimicrobial agent, tobramycin-loaded mesoporous silica particles by covering its surface via electrostatic attraction that is triggered into action upon sensing the motion of swarmer *P. mirabilis*. The copolymer chains were also conjugated to peptide ligands YVLWKRKRKFCFI-NH₂ that display affinity to Gram-negative bacteria. The copolymer-YVLWKRKRKFCFI-NH₂ binds to the lipopolysaccharides on the outer membrane of motile *P. mirabilis* and were stripped off the particle surface when the cells move away; hence releasing tobramycin into the swarmer colony and inhibiting its expansion. The materials were further in vitro release of tobramycin from pNIPAAm-co-AEMA copolymer was studied as well as antimicrobial screening and efficacy studies.

The paper is interesting, containing novelty in the application and displays a lot of work, well documented and the discussion of the obtained results is profound. Although pNIPAAm and AEMA copolymer has previously combined with mesoporous silica nanoparticles this does appear to be the first work looking at pNIPAAm-co-AEMA copolymer and mesoporous silica nanoparticles and tobramycin applied as new bacteria triggered drug release mechanism as Motion-Induced Mechanical Stripping (MIMS). However, according to my opinion, to be useful to the scientific community, it still needs some minor changes and improvements necessary before the work is suitable for publication:

Results and Discussion

- Copolymer 3 seals in payloads:

1) Line 144: mesoporous silica particles without a copolymer shell at various times...change to copolymer.

Methods

- Dye and Drug Release Experiments

2) It is unclear how the authors determine the loaded amount of tobramycin. In the Method section they claim to use UV spectrometry to quantify it - I suppose it is w% and it should be specified. I suppose that the author measured the concentration of tobramycin before and after the loading procedure, but it is not really explicit in the text.

3) Any kinetic model was used? I think its use to analyzing the data presented here is important for validity of the analysis to be adequately assessed.

4) Is there any references upon which your drug loading protocol was based?

After drug loading did samples remain hydrated? If it wasn't what was the drying protocol?

5) Why was the incorporation with toluidine blue (TB) carried out? It is not explicit in the text.

6) How does pNIPAAm-co-AEMA copolymer in combination with tobramycin alleviate these concerns considering toxicology or cytotoxicity for the use in human? Even with the toxicity test with the bacteria presented in the item: Copolymer 3-coated silica particles containing drug inhibits swarmer *P. mirabilis* colony expansion on agar surface. But it is known that different cell types may have different sensitivities.

7) Additional referencing could be included for the short and long term toxicological effects of nanoparticles and nanostructures, for example: Elsaesser, A. & Howard, C. V. (2012) Toxicology of

nanoparticles. *Advanced Drug Delivery Reviews* 64, 129-137; Psarra E, König U, Ueda Y, Bellmann C, Janke A, Bittrich E, et al. Nanostructured biointerfaces: nanoarchitectonics of thermoresponsive polymer brushes impact protein adsorption and cell adhesion. *ACS Appl Mater Interfaces* 2015; 7: 12516–29.

- Colony expansion and proliferation tests and Wide-field fluorescence microscopy setup

8) Is there any references upon which your carrying out the protocol of this test was based?

In my opinion, after these improvements, the paper should be worth being published in the nature communication.

Reviewer #2 (Remarks to the Author):

This manuscript is a well written, quite interesting scientific report. The idea of using a loosely bound polymer to entrap an antibiotic in the pores of mesoporous silica to be released upon the polymer coating attaching more strongly to the mobile bacteria and releasing the drug, killing the bacteria is very clever. The claims of this paper that the authors have designed and produced an intelligent drug release system are well supported by the seemingly reproducible results of an I-DRS that is responsive to mobile gram negative bacteria.

1. The authors use a pNIPAAm derivative polymer to retain the drug in the pores. As is well known, this is a temperature responsive polymer. Is there a reason (other than the stated reasons on page 5) that the authors are using a environment stimulated polymer? Does the temperature sensitivity of pNIPAAm contribute to the observed results? There are other polymers that do not respond to changes in temperature that are "non-toxic, soluble in water and convenient (sp) to synthesize and functionalize (sp)". Does the properties of pNIPAAm contribute?

2. More explanation to how the amount of tobramycin loaded in the mesopores is needed. The authors state they estimate a loading of 200 mg of drug per gram of particles. Figure 3B indicates they release upwards of 90% of the total drug in 100 hours. Tobramycin has a number of primary amines that one would think would electrostatically interact strongly with the surface silicates, thus staying in the pores. More information is needed describing quantification methods. Determining drug loading by measuring and analyzing release curves may not be the best method.

3. Can the authors elaborate on the methods on how the particles were evenly distributed on half the petri dishes?

Reviewer #1:

1) *Line 144: mesoporous silica particles without a copolymer shell at various times...change to copolymer.*

Response: We have made the change in the revised manuscript.

2) *It is unclear how the authors determine the loaded amount of tobramycin. In the Method section they claim to use UV spectrometry to quantify it - I suppose it is w% and it should be specified. I suppose that the author measured the concentration of tobramycin before and after the loading procedure, but it is not really explicit in the text.*

Response: The method of determining the loaded amount of Tob is as follows: A weighed amount of silica particles (~ 5 mg) were loaded with tobramycin by stirring an aqueous solution (~ 500 μL) of tobramycin (400 mg mL^{-1}) with the particles for 3 days. The drug loaded particles were recovered by centrifugation, washed with DI water (0.5 mL), freeze dried and weighed again. The loading amount was obtained by the difference in mass of silica particles before and after loading with tobramycin. For our experiment, the amount of tobramycin loaded is $0.234 \pm 0.006 \text{ mg}$ of drug/1 mg of silica particles. The maximum amount of drug released, according to Fig. 3B, is ~ 0.2 mg of drug/1 mg of particles which is comparable to the amount of drug loaded. We have included the experimental details (page S7, SI) and the amount of drug loaded (page 8, main text) in the revised manuscript. UV spectrometry was not used to determine the amount of drug loaded since the amount is significantly smaller than the amount of drug used in the sample preparation, and the difference in the absorbance of the initial solution and supernatant after centrifugation is too small to be measured using this technique. The UV spectrometry method was used to obtain the drug release curve (Fig. 3B).

3) *Any kinetic model was used? I think its use to analyzing the data presented here is important for validity of the analysis to be adequately assessed.*

Response: We used the Higuchi model to describe the drug release kinetics.^{A1} This model, based on a diffusion controlled release of drug from a matrix system, is commonly utilized to describe the kinetics of drug release from mesoporous silica.^{A2,A3} According to the Higuchi model, the amount of drug released after time t is given by Q :

$$Q = \sqrt{\frac{D\varepsilon}{\tau}(2C - \varepsilon C_s)C_s t}$$

where D is the diffusion coefficient of the drug in the matrix solution, ε is the porosity of the matrix, τ is the capillary tortuosity factor, C is the total amount of drug in the matrix and C_s is the solubility of drug in the matrix solution. Therefore, for a diffusion controlled release of drug, Q displays a linear relationship with $t^{1/2}$.

Fig. A1 shows a plot of Q vs. $t^{1/2}$ for the release of tobramycin based on Fig. 3B (main text). We note from Fig. A1 that the antibiotic released from silica particles follows a two-step process; a fast release within the first 2 h followed by a slower release. For both steps, the Q vs. $t^{1/2}$ plot can be described by a linear relationship, suggesting a diffusion controlled kinetics for the release of tobramycin. The last point in Fig. A1 at $t^{1/2} = 9.8 \text{ h}^{1/2}$ shows slight deviation from the linear fit since it is close to the time region when the amount of drug released has reached steady-state. The first (fast) step is likely due to the release of drug molecules that are either weakly bound to the silica pore surfaces or not efficiently encapsulated within the matrix,^{A4} whereas the second (slower) release step is due to drug molecules that interact relatively stronger with silica due to the amine groups on tobramycin. Another plausible explanation of the two-step process could be due to different dissolution rates of silica.^{A2} We have included the above information (page S12 in SI and page 9 in main text) in the revised manuscript.

Fig. A1. Amount of tobramycin released vs. square-root of time. The correlation coefficients of the linear fits for the first and second steps are $R_c = 0.98$ and 0.97 , respectively.

4) *Is there any references upon which your drug loading protocol was based? After drug loading did samples remain hydrated? If it wasn't what was the drying protocol?*

Response: The drug loading protocol used in our work and as discussed in point # (1) above (*i.e.*, impregnation from a solution containing the drug) is a standard method that is generally used (see ref. A2 –A4). This information and references are now included in the revised manuscript (page S7 in SI and page 19 in main text). Apart from the experiment to determine the amount of drug loading (point # (1)), for all the other experiments, after the drug has been loaded into the mesoporous silica particles, the particles were used immediately for further experiments while remaining hydrated.

5) *Why was the incorporation with toluidine blue (TB) carried out? It is not explicit in the text.*

Response: Toluidine blue (TB) is a positively charged blue dye that can be readily quantified using spectroscopic methods such as UV-vis absorbance. The strong absorbance of TB at visible wavelengths makes it a convenient indicator to quickly assess the loading ability of the dye into polymer-coated silica particles. We have therefore chosen TB as the first active material to demonstrate that copolymer **3** is able to effectively seal the dye molecules in the particles. We have now made this clearer in the revised manuscript (page 7 in main text).

6) *How does pNIPAAm-co-AEMA copolymer in combination with tobramycin alleviate these concerns considering toxicology or cytotoxicity for the use in human? Even with the toxicity test with the bacteria presented in the item: Copolymer 3-coated silica particles containing drug inhibits swarmer P. mirabilis colony expansion on agar surface. But it is known that different cell types may have different sensitivities.*

Response: Unlike its monomer, the polymer pNIPAAm has been assessed to show little cytotoxicity to cells studied by Cooperstein and Canavan.^{A5} Likewise, pNIPAAm-based copolymers have also been extensively investigated for potential biomedical applications including drug delivery.^{A6} Furthermore, poly(2-aminoethyl methacrylate) pAEMA has previously been utilized for cargo delivery to dendritic cells with little cytotoxicity effects.^{A7,A8} We have now included this information and the references in the revised manuscript (page 5 in main text). Long term toxicological examination on human models must indeed be conducted when the proposed

drug release system is coated on catheters for possible treatment of CAUTIs. The primary aim of the current study is the initial stage of demonstrating that MIMS is a functional mechanism for drug release.

7) *Additional referencing could be included for the short and long term toxicological effects of nanoparticles and nanostructures, for example: Elsaesser, A. & Howard, C. V. (2012) Toxicology of nanoparticles. Advanced Drug Delivery Reviews 64, 129-137; Psarra E, König U, Ueda Y, Bellmann C, Janke A, Bittrich E, et al. Nanostructured biointerfaces: nanoarchitectonics of thermoresponsive polymer brushes impact protein adsorption and cell adhesion. ACS Appl Mater Interfaces 2015;7:12516–29.*

Response: We have added a comment in the conclusion regarding the need for the understanding of the short and long term toxicological effects of nanoparticles and nanostructures, and included the references as suggested by the reviewer.

8) *Colony expansion and proliferation tests and Wide-field fluorescence microscopy setup - Is there any references upon which your carrying out the protocol of this test was based?*

Response: The protocol used to examine the proliferation of swarmer colonies on agar surfaces and the use of wide-field fluorescence microscopy for real-time visualization were previously used by our group to study the effects of photodynamic therapy and silica microparticles on the dynamics of coherent motion and effects on swarmer colony expansion (see Ref. A9 and A10 which are also cited in the main text).

References

A1. Higuchi, T. Mechanism of sustained-action medication. Theoretical analysis of rate of release of solid drugs dispersed in solid matrices. *J. Pharm. Sci.* **52**, 1145-1149 (1963).

A2. Andersson, J., Rosenholm, J. & Lindén, M. Mesoporous silica: an alternative diffusion controlled drug delivery system. In *Topics in Multifunctional Biomaterials and Devices Vol. I* (ed. Ashammakhi, N.) (World Scientific Publishing, Singapore, 2008).

A3. Radin, S.; Chen, T. & Ducheyne, P. The controlled release of drugs from emulsified, sol gel processed silica microspheres. *Biomaterials* **30**, 850-858 (2009).

- A4. Kim, H. -W.; Knowles, J. C. & Kim, H. -E. Hydroxyapatite porous scaffold engineered with biological polymer hybrid coating for antibiotic Vancomycin release. *J. Mater. Sci. Mater. Med.* **16**, 189-195 (2005).
- A5. Cooperstein, M. A. & Canavan, H. E. Assessment of cytotoxicity of (*N*-isopropyl acrylamide) and poly(*N*-isopropyl acrylamide)-coated surfaces. *Biointerphases* **8**, 19 (2013).
- A6. Lanzalaco S. & Armelin, E. Poly(*N*-isopropylacrylamide) and copolymers: a review on recent progress in biomedical applications. *Gels* **3**, 36 (2017).
- A7. Hu, Y.; Atukorale, P. U.; Lu, J. J.; Moon, J. J.; Um, S. H.; Cho, E. C.; Wang Y.; Chen, J & Irvin, D. J. Cytosolic delivery mediated via electrostatic surface binding of protein, virus, or siRNA cargos to pH-responsive core-shell gel particles. *Biomacromolecules* **10**, 756-765 (2009).
- A8. Ji, W.; Panus, D.; Palumbo, R. N.; Tang, R. & Wang C. Poly(2-aminoethyl methacrylate) with well-defined chain-length for DNA vaccine delivery to dendritic cells. *Biomacromolecules* **12**, 4373-4385 (2011).
- A9. Lu, S., Bi, W., Liu, F., Xing, B. and Yeow, E. K. L. Loss of collective motion in swarming bacteria undergoing stress. *Phys. Rev. Lett.* **111**, 208101 (2013).
- A10. Lu, S., Liu, F., Xing, B. & Yeow, E. K. L. Nontoxic colloidal particles impede antibiotic resistance of swarming bacteria by disrupting collective motion and speed. *Phys. Rev. E* **92**, 062706 (2015).

Reviewer #2:

1) *The authors use a pNIPAAm derivative polymer to retain the drug in the pores. As is well known, this is a temperature responsive polymer. Is there a reason (other than the stated reasons on page 5) that the authors are using an environment stimulated polymer? Does the temperature sensitivity of pNIPAAm contribute to the observed results? There are other polymers that do not respond to changes in temperature that are "non-toxic, soluble in water and convenient (sp) to synthesize and functionalize (sp)". Does the properties of pNIPAAm contribute?*

Response: Yavuz *et al.* have demonstrated that the incorporation of acrylamide (AAm) into pNIPAAm allows the lower critical solution temperature (LCST) of pNIPAAm-co-pAAm to be tuned from 32 to 50 °C, depending on the amount of AAm.^{B1} When the amount of AAm is increased, the LCST is shifted to higher temperature. For example, for pNIPAAm-co-pAAm with a molar ratio of NIPAAm : AAm = 75% : 25%, the LCST is 48.9 °C. The authors have judiciously controlled the LCST of their pNIPAAm-co-pAAm copolymer, and successfully investigated the NIR light-activated release of drug encapsulated within pNIPAAm-co-pAAm-covered gold nanocages; forming an inspiration for us to use the pNIPAAm-co-pAEMA copolymer system in our study.

We have determined the LCST of copolymer **3** (molar ratio of NIPAAm : AEMA : peptide = 60.5% : 9.8% : 28.8%) used in our study by measuring the transmission of the copolymer in PBS buffer (pH = 7.2). Fig. B1 shows separately the transmission of copolymer **3** and pNIPAAm (α,ω -bis(carboxy)-terminated) at 450 nm. The transmission (T) of completely solubilized copolymer **3** ($T \sim 94$) is slightly lower than that of solubilized pNIPAAm ($T \sim 100$) due to small absorbance of light by the peptide ligand in the former. By defining the LCST (cloud point) of the polymer to be the temperature at which the transmission (at 450 nm) decreases to 50% of the initial transmission, the LCST of pNIPAAm and copolymer **3** are determined to be 30 and 53 °C, respectively. It is pertinent to note that the LCST of copolymer **3** is significantly higher than the temperature used in our MIMS experiments (≤ 37 °C), and the copolymer does not precipitate out at the temperature range from 25 to 46 °C. Similar observation was also seen when the experiment was conducted in lower pH buffer solutions (5.8 and 2.7) where the large T values (> 90) for copolymer **3** is maintained at 40 °C. The increase in LCST compared to pNIPAAm is likely due to the incorporation of the more hydrophilic AEMA. Therefore, the temperature-dependent properties of pNIPAAm in copolymer **3** does not contribute at the temperature range used in our study. We have

included the above information in the revised manuscript (page S7 and S11 in SI and page 7 in main text)

Fig. B1. The transmission of pNIPAAm (α,ω -bis(carboxy)-terminated) and copolymer **3** in PBS buffer (pH = 7.2) at 450 nm vs. temperature.

2) *More explanation to how the amount of tobramycin loaded in the mesopores is needed. The authors state they estimate a loading of 200 mg of drug per gram of particles. Figure 3B indicates they release upwards of 90% of the total drug in 100 hours. Tobramycin has a number of primary amines that one would think would electrostatically interact strongly with the surface silicates, thus staying in the pores. More information is needed describing quantification methods. Determining drug loading by measuring and analyzing release curves may not be the best method.*

Response: The method of determining the loaded amount of Tob is as follows. A weighed amount of silica particles (~ 5 mg) were loaded with tobramycin by stirring an aqueous solution (~ 500 μL) of tobramycin (400 mg mL^{-1}) with the particles for 3 days. The drug loaded particles were recovered by centrifugation, washed with DI water (0.5 mL), freeze dried and weighed again. The loading amount is obtained by the difference in mass of silica particles before and after loading with tobramycin. For our experiment, the amount of tobramycin loaded is $0.234 \pm 0.006 \text{ mg}$ of drug/1 mg of silica particles. The maximum amount of drug released according to Fig. 3B was reported to be 0.2 mg of drug/1 mg of particles which is comparable to the amount of drug loaded. To be more precise, the kinetics release curve (Fig. 3B in main text) provides the maximum amount of drug released, and we have now made this clarification in the revised manuscript. We note that

> 85% of drug is released. We have included the experimental details (page S7, SI) and the amount of drug loaded (page 8, main text) in the revised manuscript.

Based on the Higuchi model for drug released, Tob is released from silica particles *via* a two-step diffusion controlled process; a fast release within the first 2 h followed by a slower release (Fig. S4, SI). The first (fast) step is likely due to the release of drug molecules that are either weakly bound to the silica pore surfaces or not efficiently encapsulated within the matrix,^{B2} whereas the second (slower) release step is due to drug molecules that interact relatively stronger with silica due to the amine groups on tobramycin. There is also a possibility of a small fraction the drug that is still tightly bound to the surface of the silica after a long time and remains trapped in the pores. It is worth noting that silica mesoporous particles were also previously used for the efficient loading and release of the antibiotic gentamicin, containing five amine groups (similar to Tob).^{B3,B4}

3) *Can the authors elaborate on the methods on how the particles were evenly distributed on half the petri dishes?*

Response: The copolymer-coated particles containing drug were first suspended in water and slightly vortexed. 200 μ L of homogenized suspension was dropped onto an LB-agar plate and immediately spread across the surface carefully using a L-shape spreader. The petri-dish was then allowed to sit for 30 min to allow the particles in the suspension to settle on the agar surface. This method was used previously to deposit polystyrene colloidal particles onto agar surfaces to investigate the effects of colloidal particles on the collective behavior of swarmer bacteria.^{B5} We have now made this clearer in the revised manuscript (page 22 in main text).

References

B1. Yavuz, M. S., Cheng, Y., Chen, J., Cobley, C. M., Zhang, Q., Rycenga, M., Xie, J., Kim, C., Song, K. H., Schwartz, A. G., Wang, L. V. & Xia, Y. Gold nanocages covered by smart polymers for controlled release with near-infrared light. *Nat. Mater.* **8**, 935-939 (2008).

B2. Kim, H. -W.; Knowles, J. C. & Kim, H. -E. Hydroxyapatite porous scaffold engineered with biological polymer hybrid coating for antibiotic Vancomycin release. *J. Mater. Sci. Mater. Med.* **16**, 189-195 (2005).

B3. Tamanna, T.; Bulitta, J. B. & Yu, A. Controlling antibiotic release from mesoporous silica nano drug carriers via self-assembled polyelectrolyte coating. *J. Mater. Sci.: Mater. Med.* **26**, 117 (2015).

B4. Doadrio, A. L.; Sousa, E. M. B.; Doadrio, J. C.; Pariente, J. P.; Izquierdo-Barba, I. & Vallet-Regí, M. Mesoporous SBA-15 HPLC evaluation for controlled gentamicin drug delivery. *J. Control. Release* **97**, 125-132 (2004).

B5. Lu, S., Liu, F., Xing, B. & Yeow, E. K. L. Nontoxic colloidal particles impede antibiotic resistance of swarming bacteria by disrupting collective motion and speed. *Phys. Rev. E* **92**, 062706 (2015).

Reviewer #1 (Remarks to the Author):

Reviewer #1:

1) Line 144: mesoporous silica particles without a copolymer shell at various times...change to copolymer.

Is in compliance with the requested.

2) It is unclear how the authors determine the loaded amount of tobramycin. In the Method section they claim to use UV spectrometry to quantify it - I suppose it is w% and it should be specified. I suppose that the author measured the concentration of tobramycin before and after the loading procedure, but it is not really explicit in the text

The answer is according to the requested. They have included the experimental details (page S7, SI) and the amount of drug loaded (page 8, main text) in the revised manuscript.

3) Any kinetic model was used? I think its use to analyzing the data presented here is important for validity of the analysis to be adequately assessed.

The answer is according to the requested. They have included the above information (page S12 in SI and page 9 in main text) in the revised manuscript.

4) Is there any references upon which your drug loading protocol was based? After drug loading did samples remain hydrated? If it wasn't what was the drying protocol?

The answer is according to the requested. They have included the information and references are now included in the revised manuscript (page S7 in SI and page 19 in main text).

5) Why was the incorporation with toluidine blue (TB) carried out? It is not explicit in the text.

The answer is according to the requested. They have now made this clearer in the revised manuscript (page 7 in main text).

6) How does pNIPAAm-co-AEMA copolymer in combination with tobramycin alleviate these concerns considering toxicology or cytotoxicity for the use in human? Even with the toxicity test with the bacteria presented in the item: Copolymer 3-coated silica particles containing drug inhibits swarmer *P. mirabilis* colony expansion on agar surface. But it is known that different cell types may have different sensitivities.

The answer is according to the requested. They have now included this information and the references in the revised manuscript (page 5 in main text).

7) Additional referencing could be included for the short and long term toxicological effects of nanoparticles and nanostructures, for example: Elsaesser, A. & Howard, C. V. (2012) Toxicology of nanoparticles. *Advanced Drug Delivery Reviews* 64, 129-137; Psarra E, König U, Ueda Y, Bellmann C,

Janke A, Bittrich E, et al. Nanostructured biointerfaces: nanoarchitectonics of thermoresponsive polymer brushes impact protein adsorption and cell adhesion. ACS Appl Mater Interfaces 2015;7:12516–29

Is in compliance with the requested.

8) Colony expansion and proliferation tests and Wide-field fluorescence microscopy setup - Is there any references upon which your carrying out the protocol of this test was based?

Is in compliance with the requested.

After review was verified that the answers are in agreement. The requested changes were made. The reviewed paper is interesting, well documented and the discussion of the obtained results is profound. Therefore, I recommend to accept for publication by Nature Communications.

Reviewer #2 (Remarks to the Author):

The authors have adequately responded to all reviewer comments and issues in the revision.